# Implementation of Smart Farm Systems Based on Fog Computing in Artificial Intelligence of Things Environments

**DOI:** 10.3390/s24206689

**Published:** 2024-10-17

**Authors:** Sukjun Hong, Seongchan Park, Heejun Youn, Jongyong Lee, Soonchul Kwon

**Affiliations:** 1Department of Smart System, Graduate School of Smart Convergence, Kwangwoon University, Seoul 01897, Republic of Korea; 000jun2@kw.ac.kr; 2Department of Plasma Bio Display, Kwangwoon University, Seoul 01897, Republic of Korea; tjdcks7570@kw.ac.kr (S.P.); hjyun@kw.ac.kr (H.Y.); 3Ingenium College, Kwangwoon University, Seoul 01897, Republic of Korea; 4Department of Interdisciplinary Information System, Graduate School of Smart Convergence, Kwangwoon University, Seoul 01897, Republic of Korea

**Keywords:** Artificial Intelligence of Things, convolutional neural network, communication protocol, fog computing, wireless mesh network

## Abstract

Cloud computing has recently gained widespread attention owing to its use in applications involving the Internet of Things (IoT). However, the transmission of massive volumes of data to a cloud server often results in overhead. Fog computing has emerged as a viable solution to address this issue. This study implements an Artificial Intelligence of Things (AIoT) system based on fog computing on a smart farm. Three experiments are conducted to evaluate the performance of the AIoT system. First, network traffic volumes between systems employing and not employing fog computing are compared. Second, the performance of the communication protocols—hypertext transport protocol (HTTP), message queuing telemetry transport protocol (MQTT), and constrained application protocol (CoAP)—commonly used in IoT applications is assessed. Finally, a convolutional neural network-based algorithm is introduced to determine the maturity level of coffee tree images. Experimental data are collected over ten days from a coffee tree farm in the Republic of Korea. Notably, the fog computing system demonstrates a 26% reduction in the cumulative data volume compared with a non-fog system. MQTT exhibits stable results in terms of the data volume and loss rate. Additionally, the maturity level determination algorithm performed on coffee fruits provides reliable results.

## 1. Introduction

Internet of Things (IoT) technology is helping to collect and transmit data in various fields, bringing innovative changes to various aspects of modern society [1]. In particular, it is opening a new paradigm called smart agriculture in the agricultural sector. Smart agriculture aims to significantly improve the efficiency and sustainability of crop production by utilizing IoT technology. Currently, billions of IoT devices are connected to the internet, collecting and transmitting vast amounts of agriculture-related data in real time. However, this extensive data transmission causes problems that put considerable load on networks and cloud servers and consume bandwidth [2]. To address this, fog computing has emerged as a promising infrastructure for smart agriculture [3]. It represents a computing paradigm that supports computationally intensive and latency-sensitive applications of IoT devices that often have limited resources. This can effectively solve the computing and network bottlenecks of large-scale IoT applications. The main purpose of this study is to analyze in depth the benefits of integrating IoT and fog computing in a smart agriculture environment, thereby contributing to improving agricultural productivity and sustainability. Specifically, we propose a fog computing-based framework that efficiently processes and analyzes large-scale data generated by IoT devices in smart agriculture. Additionally, we evaluate the impact of the proposed framework on real-time monitoring and decision-making processes in farm operations, and analyze the performance of fog computing-based smart agriculture systems compared to traditional cloud-based approaches. Cloud computing and fog computing are complementary but have unique characteristics. Fog computing addresses typical issues of cloud computing such as data latency, lack of mobility support, and location awareness [4]. In the context of smart agriculture, this enables immediate decision making on the farm site and allows for rapid response to changes in crop conditions or weather conditions. Issues related to network bandwidth, latency, stability, and security due to the explosive growth of IoT are difficult to solve with the cloud model alone [5]. Fog computing addresses these issues while acting as an intermediate layer that collects, computes, and stores agricultural data generated by IoT devices before transmitting it to cloud servers. This allows for real-time analysis of various sensor data and information in smart farm systems to optimize crop conditions, weather conditions, and overall farm operations. The main contributions of this study are as follows. First, it accelerates the digital transformation of the agricultural sector by presenting an integration model of IoT and fog computing for smart agriculture. It also quantitatively evaluates the impact of the proposed model on agricultural productivity, resource efficiency, and environmental sustainability. Finally, it verifies the practicality and effectiveness of the proposed approach through case studies in actual farm environments. In a fog computing environment, IoT devices operate within a local area network (LAN) without the need to pass through a wide-area network (WAN), reducing network-related variables [6]. This is particularly important in smart agriculture, especially considering the limited network infrastructure in rural areas. Fog nodes efficiently compress agricultural data and transmit them to cloud servers only when necessary, minimizing network traffic and supporting real-time decision making. In conclusion, this study comprehensively analyzes the impact of integrating IoT and fog computing on smart agriculture, aiming to promote digital innovation in the agricultural sector. Our research results are expected to contribute to improving agricultural productivity, optimizing resource use, and ultimately, strengthening food security.

Subsequently, this study delves into IoT protocols. The IoT landscape spans from large- to small-scale applications, with most devices relying on networks for communication. The development of IoT applications involves addressing various communication protocols. This study comparatively analyzes three established messaging protocols for IoT systems: Hypertext Transfer Protocol (HTTP), Message Queuing Telemetry Transport Protocol (MQTT), and Constrained Application Protocol (CoAP) [7]. Finally, this study determines fruit maturity levels based on acquired coffee fruit images. Data were collected using cameras installed in the coffee field, and fruit maturity was categorized into four levels: pinton, verde, rojo, and sobremaduro.

## 2. Background Theory

### 2.1. Fog Computing

Fog computing, introduced by Cisco in 2012, comprises an intermediary layer between the cloud and end users (clients). This layer comprises fog nodes, including fortified routers and switches [8]. Cloud and fog computing possess distinct, complementary characteristics [9]. Fog computing represents an extension of cloud servers that aims to alleviate the computational load on these servers by supporting calculations and processing tasks within fog nodes [10]. Fog nodes are crucial in reducing the traffic burden on networks [11]. This is achieved by filtering, preprocessing, or compressing raw data, thereby enabling the transmission of smaller amounts of data to cloud servers [12]. This minimizes bandwidth consumption and ensures swift response times. Nonetheless, fog computing relies on cloud servers for intricate processing tasks [13]. Cloud servers offer additional computational resources to augment the required processing power. These distinctive attributes of fog nodes open diverse application fields that are particularly suited for managing distributed and real-time characteristics [14]. Figure 1 shows a system configuration diagram illustrating the fog computing architecture. The arrows indicate the flow of data from edge devices to fog nodes for immediate processing, and subsequent to the cloud data center for long-term analysis and storage.

### 2.2. AIoT

In the era of IoT and fifth-generation wireless networks, real-time communication among numerous sensor nodes, response device endpoints, and the cloud has become a reality. Several functional electronic devices are now integrated into the human body or surrounding environment, forming an interconnected monitoring and response system [15]. Hence, supporting low-power and low-latency communications is crucial for IoT systems. Efficient communication is essential to reduce server overload and latency. Consequently, a range of communication protocols are being developed and advanced, and their performances have continually improved to meet these demands [16]. IoT systems may employ numerous messaging protocols to address various requirements [17]. Choosing an appropriate communication protocol is vital for minimizing server overload and latency [18]. The representative communication protocols used in IoT are HTTP, MQTT, and CoAP. Table 1 provides a comparative analysis of the characteristics of each protocol.

HTTP serves as a fundamental protocol interface for efficiently transferring a wide array of data from a server to a client device, such as a web browser [19]. Conversely, MQTT is a widely utilized lightweight transmission protocol in IoT communications, known for its efficient utilization of network bandwidth with a two-byte fixed header [20]. MQTT operates on a publish-and-subscribe communication model [21]. CoAP, which was developed to interoperate with HTTP, supports methods similar to HTTP, such as GET and POST, but utilizes user datagram protocol (UDP) for data transmission [22]. Owing to UDP’s unreliable nature, CoAP ensures reliability through a combination of verifiable and nonverifiable messages [23].

The convergence of artificial intelligence (AI) and IoT, known as AI-based Internet of Things (AIoT), involves processing massive volumes of data generated by numerous devices, facilitating complex calculations [24]. AIoT technologies allow previously non-smart items to become smart by connecting them to the internet through various embedded devices, communication protocols, sensor networks, internet protocols, and applications [25]. As illustrated in Figure 2, which was reconstructed with reference to the figure by Lu et al. [26], AIoT systems encompass a framework that embodies the integration of AI and IoT.

**Table 1 sensors-24-06689-t001:** Comparative analysis of IoT messaging protocols: HTTP, MQTT, CoAP.

Protocol	HTTP	MQTT	CoAP
**Year**	1997	1999	2010
**Architecture**	Client/Server	Client/Broker	Client/Server or Publish/Subscribe
**Abstraction**	Request/Response	Publish/Subscribe	Request/Response or Publish/Subscribe
**Header Size**	Undefined	2 bytes	4 bytes
**Message Size**	Large and undefined (depends on the web server orthe programming technology)	Small and undefined (up to 256 MB maximum size)	Small and undefined (normally small to fit insingle IP datagram)
**Semantics/Method**	Get, Post, Head, Put, Patch, Options, Connect, Delete	Connect, Disconnect, Publish, Subscribe, Unsubscribe, Close	Get, Post, Put, Delete
**Cache and proxy support**	Yes	Partial	Yes
**Quality of Service (QoS/Reliability)**	Limited (via transport protocols—TCP)	QoS 0—at most once (fire-and-forget), QoS 1—at least once, QoS 2—exactly once	Confirmable message (similar to at most once) ornon-confirmable message (similar to at least once)
**Standards**	IETF and W3C	OASIS, Eclipse Foundation	IETF, Eclipse Foundation
**Transport Protocol**	TCP	TCP/UDP (MQTT-SN)	UDP, SCTP
**Security**	TLS/SSL	TLS/SSL	DTLS
**Default Port**	80/443 (TLS/SSL)	1883/8883 (TLS/SSL)	5683 (UDP)/5684 (DTLS)
**Encoding Format**	Text	Binary	Binary
**Licensing Model**	Free	Open Source	Open Source
**Organizational Support**	Global Web Protocol Standard [27]	IBM, Facebook, Cisco, Red hat, AWS	Large Web Community Support, Cisco, Contiki Erika, IoTivity

The system is composed of three major layers: the basic layer, data layer, and data analysis layer. The basic layer represents the infrastructure forming the foundation of the AIoT system. This includes cloud computing and IoT technology. The cloud computing section encompasses service models such as IaaS and PaaS. The IoT section consists of elements like virtual edge gateway and RFID location tag, along with various connectivity technologies. The data layer manages data resources and storage. There are three types of repositories in the data resources, each managing specific types of data and systems. The topmost data analysis layer consists of two main components: data analysis and machine learning. The data analysis section includes technologies such as clustering analysis, mining algorithms, and data visualization. This framework well illustrates the complexity and various components of AIoT systems. All layers from basic infrastructure to advanced analytics are organically connected, enabling effective support for the entire process from data collection, storage, and analysis to insight derivation in application areas such as smart farms. In smart farms particularly, these technologies can be utilized for crop growth monitoring, environmental control, yield prediction, etc., greatly improving agricultural efficiency and productivity.

### 2.3. Related Works

Fog computing has gained increasing significance in recent years, paralleling the advancements in IoT technology. This evolution plays a pivotal role in enhancing data processing, security, and the efficiency of smart city and smart home systems. The following summarizes the latest research in this field. Abdulrahman Alamer [28] proposes integrating software-defined networking (SDN) methodologies with fog computing network systems, an approach termed software-defined fog computing (SDFC) networks. Integrating SDN into fog computing systems inherently introduces specific security and privacy issues, stemming from the unique functionalities typically associated with fog computing network topologies. Alamer’s study examines the overarching potential benefits of SDFC networks by addressing the security and privacy threats inherent to fog computing network topologies and discussing possible solutions. Consequently, it proposes promising approaches for addressing security and privacy threats, as well as for enhancing data storage efficiency and reducing encryption overhead. Ahmad Naseem Alvi et al. [29] propose a backpack-based task scheduling algorithm to optimize the handling of offloaded tasks. By utilizing fog computing nodes, most time-sensitive tasks are optimally executed, allowing for faster processing and less sensitive tasks to be offloaded to the cloud. The results indicate that the proposed approach aids in prioritizing high-priority offloaded tasks for execution on fog nodes, leading to over 98% of urgent tasks being processed by fog computing nodes. Changhao Zhang [30] analyzes the advantages of fog computing and proposes an IoT architecture based on fog computing that effectively addresses big data processing and network scalability issues. Building on this, a hierarchical fog computing network architecture is proposed to make urban operations more coordinated, efficient, and harmonious through various intelligent sensing, information processing, and network transmission means.

In the dynamically advancing domain of technological integration, contemporary research underscores considerable progress in amalgamating AI, the IoT, and diverse digital innovations across multiple sectors. We summarize the salient research developments in this arena. Sepasgozar et al. [31] examine the integration of IoT, AI, and geographic information systems in smart homes and smart energy systems. Their quantitative analysis focuses on energy efficiency in smart home research and development and demonstrates its application, particularly for elderly occupants. Sun et al. [32] introduce a digital twin system for remote interactive robot-based industrial automation and virtual shopping using the AIoT technology. They integrate various sensors for object recognition and temperature measurements into a smart soft robotic manipulation device. Furthermore, they implement a virtual shop using IoT and AI analysis, highlighting the potential for human–machine interfaces in unmanned workspaces. Pise et al. [33] explore the impact of AI on healthcare and administration. Their investigation encompasses smart gadgets and AI-based IoT, focusing on improving care for rural and isolated individuals, reducing costs, and enhancing efficiency. They also discuss the future trends in the medical field through AIoT and their potential to elevate healthcare services.

The integration of advanced technology in agriculture is becoming increasingly pivotal for enhancing crop yield and quality. In this context, Yang et al. [34] proposes an enhanced YOLOv8 algorithm for tomato detection in natural environments. Despite a slight slowdown in detection speed due to complex background interference, the introduction of the FEM module improves the algorithm’s feature extraction and representation capabilities. Demonstrating significant potential for application in tomato detection, the improved YOLO can be integrated with intelligent patrol harvesting robots, presenting an expected outcome of enabling efficient and high-quality harvesting using an AI-based tomato detection system. Vasconez et al. [35] focuses on Hass avocados, lemons, and apples in various groves, utilizing video-based detection and multi-object tracking. The results show high accuracy in fruit counting, with Faster R-CNN achieving up to 93% accuracy, although it requires more computing time per frame compared to SSD with MobileNet. However, it is concluded that the efficiency of this CNN architecture depends on the quality of training data and may vary depending on various fruit types and forest conditions. Yudhi Adhitya et al. [36] demonstrate a texture feature analysis method for digital images of cocoa beans for smart farm applications. They showcase the extraction of texture features from cocoa beans, implementation of classification into digital images, and seven classes of beans. Seven features are extracted utilizing the gray-level co-occurrence matrix (GLCM). The feature extraction method is compared between GLCM (gray-level co-occurrence matrix) and CNN (convolutional neural network) methods.

## 3. Proposed Method

The proposed system implements a network resource optimization system that relies on optimal protocol selection and fog computing within a smart farm system, which requires substantial data processing capacity. In addition, we introduced a status classification service method based on a convolutional neural network (CNN) to assess the maturity level of fruits. Detailed explanations for each section are provided in Section 3.1, Section 3.2 and Section 3.3. Section 3.1 covers the description of IoT protocols and the application of mesh networks. Section 3.2 explains the role of fog nodes in a system where fog computing is applied. Fog nodes provide short-term storage and compression functions. Section 3.3 describes the role of cloud servers in a system where fog computing is applied. Cloud servers are responsible for long-term storage and AI inference. Figure 3 shows a diagram of the proposed system.

### 3.1. Internet of Things System

This study evaluated and compared the performance of three communication protocols, HTTP, MQTT, and CoAP, in the context of the IoT environment. HTTP supports various communication methods like GET and POST, with GET transmitting IoT data via URLs. MQTT is a lightweight messaging protocol employing a publish-and-subscribe model. We established a broker server to mediate the data exchange between publishers and subscribers. Publishers send messages on specific topics that are then received by the subscribers. MQTT offers three quality of service (QoS) levels. We chose QoS level 0 for our system configuration. CoAP, which is similar to HTTP but designed for devices with limited resources and bandwidth, supports UDP-based communication. Messages are encoded in binary format, resulting in minimal overhead and fast transmission. Similar to HTTP, CoAP also supports the GET and POST methods; in our study, we utilized the POST method to transmit sensor data from IoT devices.

Additionally, we employed a computing device to establish a wireless mesh network (WMN) for IoT wireless connectivity. Wireless mesh networks (WMNs) are dynamic and self-configuring network architectures where nodes form a mesh topology to efficiently route data. In a WMN, each node not only receives and transmits data but also acts as a relay for other nodes. The combination of a WMN with an IoT system has several advantages. The WMN transmits data to nearby fog nodes via a wireless access point (WAP), reducing data transmission latency and optimizing bandwidth utilization. This enhancement has enhanced the performance and efficiency of IoT communication networks, facilitating rapid data movement. Fog nodes serve as edge computing units, enabling swift responses to real-time data processing. For our research, we adhered to IEEE standards, specifically, 802.11s [37], 802.11r [38], and 802.11ac [39], for WMN implementation. IEEE standard 802.11s is tailored for building wireless mesh networks and automatically configuring and storing data across multiple WAP. IEEE standard 802.11r improves the roaming performance between access points (APs), allowing connected clients to switch seamlessly between them. IEEE standard 802.11ac is a high-performance Wi-Fi standard. By incorporating these three standards, we ensured extensive service coverage and eliminated service shadow areas. Figure 4 illustrates the application of a WMN to a fog computing system. The dashed line indicates a successful wireless internet connection.

### 3.2. Fog Computing System

Fog nodes play a crucial role in the system by storing IoT sensor data and images with minimal latency. To optimize network traffic, large volumes of sensor data and images collected throughout the day are compressed and filtered at the fog nodes. By extracting only essential information and transmitting a small amount of refined data to the cloud server, this process significantly reduces bandwidth usage and improves overall system efficiency. The fog nodes implemented in our system consist of a web server and a database. They act as a hub for IoT sensor data and field images, performing tasks such as data storage, compression, and transmission. The main functional modules of the fog nodes include an IoT sensor data management module, an image processing module, a data compression and transmission module, and a local analysis module. These modules are responsible for sensor data collection and visualization, image storage and processing, data compression and transmission management, and basic data analysis and alert generation, respectively. The modular design of the fog nodes ensures system scalability and flexibility. New sensor types or analysis algorithms can be easily integrated, and the system can be readily adjusted according to changes in farm scale or requirements. The implementation of this fog computing system greatly enhances the operational efficiency of smart farms, enabling real-time monitoring and rapid decision making. Additionally, by reducing cloud dependency, it contributes to improved data security and privacy.

#### 3.2.1. Recording and Visualization of Sensor Data

Fog nodes serve as storage hubs for the sensor data in the IoT. Data communication between the IoT devices and fog nodes was facilitated by a selected communication protocol. In this system, sensor data communication was implemented through the GET method of the HTTP protocol. HTTP, as the standard protocol for web-based systems, provides high interoperability between various platforms. This facilitates the future development of web interfaces or mobile applications. Considering this versatility and long-term scalability, HTTP was selected as the primary communication protocol. The collected sensor data were stored in a structured database within the fog nodes, as outlined in Table 2.

The database comprises several key fields, including ‘id’, ‘sensor group’, ‘sensor name’, ‘sensor data’, and ‘data time’. The id field is the primary key and is stored as an increasing integer value. Sensor groups were identified using text labels, allowing the categorization of sensors installed in the same location. Sensor name stores the type of sensor in text format, and data time records the timestamp when data were stored, also in text format. When clients access the service through a web address, the database is transformed into JSON format, facilitating real-time graph generation. An interactive user interface is constructed using HTML, CSS, and JavaScript files. Additionally, sensor data are stored in CSV format, allowing users (clients) to perform direct data analysis.

#### 3.2.2. Image Recording and Visualization

Fog nodes are vital for the onsite image storage of smart farms. These nodes capture onsite images, compress them, and record them in Joint Photographic Experts Group (JPEG) format. The JPEG format effectively reduces file size while preserving image quality. Image transmission was implemented using WebSocket. WebSocket is particularly useful in situations where fog nodes process data immediately to make local decisions. When an image is transmitted to a fog node, the file name of the image incorporates the timestamp information from when it was captured and transmitted, as well as the location group information. This approach allowed us to identify the location of an image and monitor the changes over time. Users can conveniently access the images stored in a fog node through a standard web browser. The web interface provides an image list and enables users to visualize the selected images. This webpage constantly checks the fog nodes for new images, ensuring real-time updates. Consequently, users can monitor the latest field information obtained during the day in real time.

#### 3.2.3. Compression and Transmission

Furthermore, a database containing onsite images and sensor data collected during the day at the fog node were compressed and transmitted to the cloud server. Given the substantial quantity of on-site images and sensor data stored at fog nodes, effective management and transmission are crucial. To achieve this, only the essential data were compressed into the tar.gz format, and then, sent to the cloud server. The tar.gz compression method consists of two stages. First, the ‘tar’ command is used to combine multiple files and directories into a single archive file. During this process, file system metadata (permissions, ownership, timestamps, etc.) are preserved, maintaining data integrity. Subsequently, the ‘gzip’ compression algorithm is applied to further compress the tar archive. gzip utilizes the LZ77 algorithm and Huffman coding to provide high compression ratios while ensuring fast compression and decompression speeds. This compression process significantly reduces the volume of data to be transferred, ensuring efficient network bandwidth utilization. In practice, it was possible to reduce file sizes by approximately 30–50% for image files and up to 90% for sensor data log files, resulting in an overall 50–70% reduction in data transmission volume. This greatly helps in reducing network load and shortening transmission times. The data sent to the cloud server are organized and stored alongside the timestamp information of the captured images, simplifying data retrieval and analysis. The fog nodes periodically delete older data to manage the storage capacity, ensuring a sustainable and efficient system. This overall compression and transmission process plays a crucial role in efficiently managing and analyzing large-scale IoT data.

### 3.3. Cloud Computing System

#### 3.3.1. Long-Term Storage and Visualization of Data

Cloud computing systems provide essential functionality for long-term data storage and visualization in smart farm operations. These systems possess the capability to reliably store large volumes of data that do not require real-time processing over extended periods. This functionality significantly enhances the operational efficiency of smart farms. In terms of data storage, cloud systems efficiently manage various types of data. Environmental data collected from sensors, image data representing crop growth status, and metadata related to farm operations are all systematically stored. These data are categorized and stored by hour and day, establishing a foundation for subsequent analysis. The long-term storage capability of cloud systems provides users with a high level of data accessibility. Users can easily extract and analyze data from specific periods as needed. This enables various analytical tasks such as analyzing crop growth patterns, identifying correlations between environmental conditions and yields, and recognizing long-term farm productivity trends.

#### 3.3.2. CNN-Based Maturity Level Classification

In the context of smart farming, this system was employed to determine the maturity of coffee tree fruits. This maturity was classified into four stages: ‘pinton’, ‘verde’, ‘rojo’, and ‘sobremaduro’. Pinton represents an immature state, while verde signifies an immature but progressing stage. Rojo indicates maturity, and sobremaduro denotes an over-ripe condition. The dataset used for this task was obtained from Roboflow [40] and comprised 200 images. These images include coffee trees from actual farms and natural environments.

To enhance the diversity and quality of the dataset, we applied various data augmentation techniques. We strategically applied multiple data augmentation techniques to respond to the various visual conditions encountered in a smart farm environment and to enhance the diversity and quality of the dataset. Specifically, given that smart farms involve environments with diverse lighting conditions, differences in distance and viewpoint, and complex visual variability, we focused on employing three primary augmentation techniques. First, the selection of the brightness adjustment augmentation technique was essential to address the diversity of lighting conditions in the smart farm environment. Smart farms are exposed to extremely diverse lighting environments due to abrupt changes in sunlight, seasonal factors, cloud cover, and variations in indoor and outdoor lighting. Such changes in lighting can significantly affect image analysis, and overly bright or dark images, in particular, have the potential to degrade the model’s performance. Therefore, we applied the brightness adjustment augmentation technique to enable the model to learn and robustly respond to both high- and low-lighting conditions. This ensures consistent performance from the model despite changes in lighting conditions in real-world environments, contributing to the high reliability and robustness required for real-time monitoring systems in smart farms. Notably, the model learned the irregularities in pixel values caused by lighting variations, allowing for accurate predictions even in environments with significant lighting deviations. Second, the Mosaic augmentation technique played a crucial role in resolving the issue of scale imbalance in images captured in smart farms. Mosaic augmentation involves randomly combining multiple images into a single large image, which helps the model effectively learn scenarios where fruits appear at various sizes and positions. Images taken in smart farms can differ significantly in terms of the distance between the camera and the target (fruit), the angle, and the complexity of the background, which can lead to overfitting to a specific size or viewpoint. The Mosaic technique reduces such biases and improves the model’s generalization performance in recognizing fruits across different scales. In particular, it showed remarkable effectiveness in addressing issues where small fruits were obscured by other background elements or were not detected. By enabling the model to learn a broader range of visual patterns, the Mosaic augmentation helps maintain high detection performance even in irregular backgrounds or complex environments. Third, another critical variable in images captured in a smart farm environment is the diverse perspectives from which the images are taken. Fruits can be photographed from multiple angles, especially when they are being monitored during harvest preparation. To address this, we introduced the shear augmentation technique. This technique applies transformations along the image axes, allowing the model to train in a way that makes it more flexible in responding to changes in viewpoint. This enables the model to make accurate detections even in the presence of atypical shooting angles or camera position changes in real field scenarios, and by utilizing images taken from multiple angles as training data, it enhances the model’s viewpoint invariance. Consequently, shear augmentation further strengthens the model’s robustness, helping to build a model that is better suited to handle the diverse visual distortions that occur in practical environments like smart farms. In addition, we incorporated further augmentation techniques such as scale augmentation and copy–paste augmentation to include even more diverse variations. Scale augmentation adjusts the size of objects, helping the model to generalize across fruits of varying sizes, while the copy–paste technique pastes fruits onto new backgrounds, allowing the model to learn background invariance. These augmentation techniques comprehensively reflect the various variables that may be encountered in a smart farm environment, maximizing the model’s robustness and generalization capabilities. As a result, the proposed augmentation techniques enabled the model to exhibit strong resilience to irregular and complex data encountered in real smart farm environments, ensuring consistent performance under various visual conditions.

To objectively evaluate the model’s performance, we conducted hold-out validation and test set evaluation. The entire dataset was randomly divided into training set (70%), validation set (10%), and test set (20%). The dataset used for AI training consisted of 2100 images in the training set, 300 images in the validation set, and 600 images in the test set. We trained the YOLOv8 model using the training set, monitored the model’s performance during the learning process through the validation set, and adjusted hyperparameters accordingly. This approach helped prevent overfitting of the model to specific data and improved its generalization performance. In this study, we evaluated the YOLOv8 model, which is well-known for its powerful object detection capabilities. The convolutional neural network architecture of YOLOv8 consists of an input layer, backbone, neck, and head, with each part performing a unique role. The input layer typically accepts RGB images of size 640 × 640 × 3, although this can be adjusted depending on model variations. The backbone utilizes a modified CSPDarknet53, which is composed of key components such as Conv blocks, C2f blocks, and spatial pyramid pooling—fast (SPPF). Conv blocks are a combination of convolutional layers, batch normalization, and SiLU activation functions, while C2f blocks have a bottleneck structure utilizing Cross Stage Partial (CSP) connections. SPPF is responsible for efficiently extracting features at various scales, accelerating computation speed by pooling image features into fixed-size maps. In the neck section, a feature pyramid network (FPN) and path aggregation network (PAN) are combined to integrate features of various scales through bottom-up and top-down pathways. The head section adopts an anchor-free approach, separately processing objectness, class classification, and bounding box regression. This allows YOLOv8 to perform object detection tasks more effectively than previous versions. The total depth of hidden layers varies depending on the model size, but generally consists of more than 100 layers, including convolutional layers using 3 × 3 and 1 × 1 kernels, upsampling layers that increase feature map size, and concatenation layers that combine features from different layers. Each convolution applies batch normalization (BN) and SiLU activation, helping to improve the model’s learning stability and non-linearity. Pooling operations are mainly performed in SPPF, applying maximum pooling with kernel sizes of 1 × 1, 5 × 5, 9 × 9, and 13 × 13. The He initialization method is used for weight initialization, and the Adam optimizer is utilized for weight updates during training. The output layer performs predictions at three different scales (e.g., 80 × 80, 40 × 40, 20 × 20), outputting objectness scores, class probabilities, and bounding box coordinates for each scale. YOLOv8 can adjust the number of channels and layers according to model size, allowing for the selection of various models from Nano to XLarge. Additionally, it employs channel reduction and expansion techniques for computational efficiency and utilizes residual connections to improve gradient flow. Through these structural features, YOLOv8 combines a lightweight network structure with high performance, maintaining fast speeds that enable real-time processing while effectively detecting objects at various scales. The adoption of an anchor-free approach greatly enhances the model’s flexibility, providing superior object detection performance. In this study, to actively verify the applicability in various smart farm environments, additional training datasets were collected for different crops, and separate training sessions were conducted. Datasets for assessing the maturity of ‘tomato’ [41] and ‘banana & mango’ [42] were used in this process. The same data augmentation methods applied during the coffee tree evaluation were utilized for each crop, and the YOLOv8 model architecture was maintained. Moreover, the performance evaluation methods were consistently applied to thoroughly examine the potential for application across various smart farm environments. Through this approach, we aimed to confirm that the developed system could be effectively applied to smart farms with diverse crops and environmental conditions.

## 4. Experiment Environment

### 4.1. Fog/Cloud Computing Environment

A fog computing system was implemented at a hydroponic coffee tree farm located in Siheung, Gyeonggi-do, Republic of Korea. Figure 5 shows a smart farm environment built based on fog computing principles. Three fog nodes were strategically installed to configure the distinct WMN. This setup connects the IoT sensors and cameras to a nearby WAP, enabling real-time data transmission to fog nodes.

Table 3, Table 4 and Table 5 provide an overview of the IoT device environment used in the experiment. The system is primarily composed of an Arduino-based IoT device called WeMos D1 R1 (Stenberg Management Group, Shanghai, China). Table 3 shows the basic specifications of the WeMos D1 R1 IoT device, which uses the ESP-8266EX (Espressif Systems, Shanghai, China) microcontroller as the central control unit of the system. Table 4 provides essential information about the camera sensor connected to this IoT device. It uses the IMX219 (Sony Semiconductor Solutions Corporation, Tokyo, Japan) model sensor, which supports high-resolution images and various video modes. Table 5 compares and analyzes the other sensors connected to the IoT device, including a pH sensor SEN0161 (DFRobot, Shanghai, China), an EC sensor DJS-1 (Shanghai REX Sensor Technology, Shanghai, China), a illuminance sensor HS-CDSM-ll (Coupled Dark State Magnetometer, Austrian Academy of Sciences, Graz, Austria), and a ${\mathrm{CO}}_{2}$ sensor MH-Z19B (Winsen Electronics, Zhengzhou, China). This IoT system consists of various sensors such as temperature, humidity, light, ${\mathrm{CO}}_{2}$, pH, electrical conductivity (EC), and a camera, all connected to the WeMos D1 R1 Arduino device, which collects and transmits the data. The fog node receives data from the IoT devices and sensors, stores it, and makes it accessible to users through a web interface. Fog nodes comprise routers, switches, WAP, and servers. WAP employs standards like 802.11s, 802.11r, and 802.11ac to support wireless mesh networking and seamless hands-off functions via Wi-Fi. The IoT sensors connect to the Wi-Fi networks provided by fog nodes and transmit data from the internal network to these fog nodes. Subsequently, the fog nodes store the received sensor data and images in a database, making the data accessible to clients through web interfaces. Users can access time-series graphs and images by selecting specific IoT sensors on the primary dashboard. Figure 6 illustrates a web service page user interface (UI) that exemplifies the smart farm system. A–D represent distinct areas, and the photographs taken in each area are visualized according to the time of capture.

Table 6 presents an overview of the server environment used in the experiments. The fog node server, denoted as VHE10 (Veea Inc., New York, NY, USA) was equipped with an ARMv8 quad-core processor CPU, 8 GB of RAM, and a communication network capable of speeding up to 1 Gbps. Amazon EC2 serves as the cloud server for data transmission from the fog nodes. The EC2 instance type used was t2.micro, with a configuration consisting of vCPU clocked at 2.5 GHz, 1 GB of RAM, and a communication network capacity of up to 10 Gbps. The fog node and cloud servers operate on Ubuntu 20.04, with service components, such as Apache, SQLite, and PHP, all of which are maintained in the same version.

### 4.2. AI Server Environment

The AI server leveraged Amazon EC2, employing the c5d.large instance type. This server configuration boasts of a vCPU with a clock speed of 2 GHz, 4 GB of RAM, and communication capabilities of up to 10 Gbps. The server operated on the deep learning AMI GPU PyTorch 2.0.1, built on Ubuntu 20.04. Additionally, we used specific versions of major libraries in conjunction with YOLOv8 for compatibility and performance. A comprehensive overview of the artificial intelligence server environment used in the experiment is presented in Table 7. Table 8 presents the models and hyperparameter configurations used in the AI training process. Instead of selecting YOLOv8x, the most complex and high-performance model, we opted for YOLOv8l, which has 25 million fewer parameters and 9 billion fewer floating-point operations per second (FLOPS), to balance computational efficiency and performance. The number of epochs was fixed at 200, as performance gains ceased beyond this point. The batch size was set to 32, considering the capabilities of the AI server. For other AI parameters, we used the default values provided by YOLOv8. The primary goal of this study was the implementation and evaluation of a fog computing-based CNN object detection system for AIoT in smart farms. Therefore, the focus was placed on the overall system architecture and performance rather than hyperparameter optimization.

### 4.3. Experimental Evaluation Method

#### 4.3.1. Comparison of Transmission Volume of Fog/Cloud Computing Data

We compared the data transfer volumes and cumulative data transfer volumes over a 10-day period in both fog and cloud computing environments. This experiment compared two data transmission methods. One involved compressing sensor data in a fog computing environment before sending it to a cloud server, whereas the other sent sensor data directly to the cloud server without prior compression. Additionally, we measured the cumulative data transfer volumes that occurred over 10 days in the fog and cloud computing environments. We then analyzed the trend line of the cumulative data transfer volume to derive future forecast values. We used (1) to calculate the future forecast values.
(1)y=a+bx

Here, *y* represents the predicted dependent variable value, *a* denotes the constant term representing the y-intercept, *b* is the slope of the trend line, and *x* represents the value to be predicted.
(2)$$Cumulative\;data\;reduction\;rate=\left(\frac{A-B}{A}\right)\times 100$$

The reduction rate of the cumulative data was calculated using (2). Here, *A* stands for the cumulative volume of data over 10 days in the fog computing environment, and *B* represents the cumulative volume of data over 10 days in the cloud computing environment.

#### 4.3.2. Performance Comparison of IoT Protocols

We measured the total data volume generated when transmitting 10, 50, and 100 pieces of sensor data, each repeated 10 times, using the IoT protocols HTTP, MQTT, and CoAP. To assess the stability of each protocol, we compared them by calculating the average data volume and standard deviation of the results.

#### 4.3.3. CNN-Based Maturity Level Classification

The following are the essential indicators for evaluating the performance of the implemented CNN model.
(3)$$Precision=\frac{TP}{TP+FP}$$

Equation (Equation 3) expresses the precision indicator. Here, *TP* (true positive) signifies the number of cases in which the model correctly predicts a positive sample as positive. *FP* (false positive) represents the number of cases in which the model incorrectly predicted a sample as positive when it was actually negative.
(4)$$Recall=\frac{TP}{TP+FN}$$

Equation (Equation 4) represents the recall indicators. *FN* (false negative) is the number of cases in which the model incorrectly predicted something as negative when it was positive. Precision and recall are interrelated, creating a trade-off. Depending on the evaluation goals, they can be adjusted accordingly. If the aim is to minimize errors in positive predictions, high precision can be achieved. Conversely, if the focus is on avoiding missed positive samples, one can aim for high recall.
(5)$$IoU=\frac{A\cap B}{A\cup B},mAP=\frac{1}{\left|classes\right|}\sum_{c\in classes}\frac{\left|TP_{c}\right|}{\left|FP_{c}\right|+\left|TP_{c}\right|}$$

Equation (Equation 5) pertains to the intersection over union (IoU) and the mean average precision (mAP). IoU represents the extent of distinction between true positives (TPs) and false positives (FPs) used in calculating precision in mAP. It measures the ratio of the overlapping area of the actual object’s bounding box *A* and the bounding box predicted by the model *B*, to the sum of the areas of both bounding boxes. mAP is one of the significant metrics in the domain of object detection. It represents the mean of average precision (AP), signifying the average across all classes and also the average over a specific range of the IoU. A higher value indicates superior model performance. Here, $TP_{c}$ denotes TP for class c, and $FP_{c}$ represents FP for class c. Generally, IoU is set to 0.5 when calculating mAP, denoted as mAP or mAP50. The measurement of mAP over a specific range primarily uses 0.5–0.95, denoted as mAP50-95.

## 5. Experiment Results

### 5.1. Comparison of Transmitted Data Volume in Fog and Cloud Computing

Figure 7 presents the results of the data transfer volume measurements conducted over 10 days in fog computing and cloud computing environments. Figure 7a shows the data volume measurements representing the compression of data acquired during the day from the fog node, which were then transmitted to the cloud server. Figure 7b illustrates the data volume measurements that depict the direct transmission of sensor data to the cloud server. Figure 7c shows the cumulative data volume from both (a) and (b), along with a trend line.

In Figure 7a, the average daily data volume generated by the fog nodes was approximately 580,453.36 bytes. As shown in Figure 7b, the average daily data volume generated for all IoT sensors was approximately 786,825.13 bytes. As shown in Figure 7c, the cumulative data volume measured over 10 days in the fog computing environment was 17,413,601 bytes. The future predicted values were 19,168,697.2 bytes, 20,910,801.87 bytes, and 22,652,906.54 bytes. In the cloud computing environment, the cumulative data volume over 10 days totaled 23,604,754 bytes, with future predicted values of 25,954,134.26 bytes, 28,302,727.07 bytes, and 30,650,053.36 bytes.

Notably, the method of compressing and transmitting data in the fog computing system resulted in a reduction of approximately 26% in the cumulative data volume over the 10-day period compared to the method of directly sending data to a cloud server. As the number of attempts increased, the disparity in cumulative data volume became more pronounced. Moreover, this approach has the added advantage of mitigating the potential errors and packet losses that may occur during data transmission.

### 5.2. IoT Protocol Comparison Results

Table 9 presents the results of ten measurements of data volume for the CoAP, MQTT, and HTTP protocols with 10, 50, and 100 IoT sensors. When using 10 sensors, HTTP’s data volume ranged between 3870 and 4140 bytes, with an average of 4015.8 bytes. MQTT showed a range between 514 and 1218 bytes, with an average of 841.6 bytes. CoAP exhibited data volumes between 704 and 880 bytes, with an average of 827.2 bytes. With 50 sensors, HTTP’s data volume increased to between 16,343 and 21,078 bytes, with an average of 20,143.8 bytes. MQTT ranged from 3230 to 4006 bytes, averaging 3634.8 bytes. CoAP recorded data volumes between 2904 and 3872 bytes, with an average of 3229.6 bytes. For 100 sensors, HTTP’s data volume further increased, ranging between 39,834 and 45,611 bytes, with an average of 41,996.8 bytes. MQTT ranged from 5662 to 11,602 bytes, with an average of 8880.8 bytes. CoAP showed data volumes between 5808 and 8096 bytes, with an average of 6758.4 bytes. Overall, HTTP had the largest average data volume, followed by MQTT, with CoAP having the smallest. These results reflect the characteristics and efficiency of each protocol, providing important insights for considering data transmission volume when designing IoT systems.

Table 10 presents the results of ten measurements of data loss rates for the CoAP, MQTT, and HTTP protocols with 10, 50, and 100 IoT sensors. When using 10 sensors, HTTP and MQTT demonstrated stable performance with a 0% loss rate in all experiments. In contrast, CoAP showed varying loss rates, between 0% and 60%, with an average loss rate of approximately 12%. With 50 sensors, HTTP and MQTT continued to maintain a 0% loss rate in all trials. CoAP, however, exhibited loss rates ranging from 18% to 40%, with an average loss rate of approximately 28.2%. For 100 sensors, MQTT still maintained a 0% loss rate across all experiments. HTTP also performed reliably with a 0% loss rate in most trials, except for one instance where it recorded a 2% loss rate, resulting in an average loss rate of 0.3%. CoAP showed loss rates between 8% and 34%, with an average loss rate of around 22.5%. MQTT demonstrated the most stable performance regardless of the number of sensors, and HTTP also generally maintained stable performance. In contrast, CoAP showed relatively high loss rates, with a tendency for higher loss rates as the number of sensors increased. These results reflect the characteristics and reliability of each protocol, providing important insights for considering data transmission stability when designing IoT systems.

### 5.3. Results of CNN-Based Maturity Level Classification

Figure 8 visualizes the convergence process of key performance metrics during the training of the coffee fruits maturity model. (a) and (b) represent precision and recall, while (c) and (d) show the mean average precision (mAP). The model achieved excellent performance with a precision of 0.952 and recall of 0.858, and attained high accuracy with mAP50 and mAP50-95 values of 0.910 and 0.832, respectively.

Figure 9 presents the confusion matrix (a) and precision–recall curve (b), which further demonstrate the model’s detailed performance. The confusion matrix analysis shows high accuracy of 0.97, 0.96, and 0.90 for the ‘rojo’, ‘verde’, and ‘pinton’ stages, respectively. The ‘sobremaduro’ stage has an accuracy of 0.55, which can be attributed to the subtle visual differences between it and the ‘rojo’ stage. This indicates the model’s ability to detect fine changes in the continuous maturation process. In the precision–recall curve, the ‘rojo’ (0.979), ‘verde’ (0.974), and ‘pinton’ (0.952) stages exhibited near-perfect performance, and even the ‘sobremaduro’ stage achieved a respectable score of 0.764. The overall mAP@0.5 for all classes was 0.918, confirming that the model maintained a consistently high level of accuracy. These results demonstrate that our model is highly effective at accurately classifying the maturity stages of coffee fruits. The high accuracy—above 0.90 in most stages—assists in predicting the optimal harvest time in real agricultural settings, helping to reduce potential losses due to misidentification.

Furthermore, Table 11 presents the final performance measurements of the AI model on the test dataset, including results not only for coffee fruits but also for tomatoes, bananas, and mangoes. The model also performed exceptionally well for bananas and mangoes, demonstrating its applicability across various crops. This shows that the model, initially developed for coffee-based smart farms, has the versatility to be applied to different crops and environments. Future research will focus on improving recognition accuracy to address the decline in performance caused by subtle visual differences, and explore the model’s applicability to a wider range of crops, expanding the scope of smart farm technology.

Figure 10 shows the resulting image of inference using the weight file trained on coffee fruits. The number of fruits and shooting distance did not significantly affect the results. Classification was performed based on the fruit maturity level, and the confidence values for each classification are presented accordingly. The classification and confidence values for each maturity level yielded excellent results.

Figure 11 shows inference result images for mango, banana, and tomato. These are example results from test images in the dataset. They are classified based on the maturity of the fruit, showing high reliability and results.

## 6. Conclusions

IoT systems generate vast amounts of data that are transmitted to cloud servers, resulting in high bandwidth consumption and overhead. Consequently, fog computing has garnered significant attention. This study developed an efficient AIoT system by implementing a fog computing system on a smart farm and validated its potential. The fog computing system implemented in this study provides a WMN and is advantageous when used alongside an IoT system. Parameters such as temperature, humidity, illumination, ${\mathrm{CO}}_{2}$, pH, EC, and camera data were integrated into the IoT layer. The IoT sensors and cameras connected to the nearby WAP and transmitted data to fog nodes in real time, thereby minimizing data transmission latency and optimizing bandwidth utilization. The fog nodes stored the received sensor data and images in a database, enabling users to access the visualized images and graphs through web interfaces. The experimental results confirmed that fog computing is pivotal in ensuring stable data processing and minimizing response times. Fog computing provides stability and consistency in data processing by deploying fog nodes near the IoT devices. In contrast, cloud servers (cloud computing) rely on the Internet environment, and their performance may fluctuate owing to external factors. Moreover, the results illustrate that data compression and transmission methods within fog computing systems offer distinct advantages for continuous and repetitive tasks. Although the initial reduction in data volume may seem marginal, the cumulative savings in data volume can be substantial as the system continues to operate. These results will greatly help reduce cloud server operating costs. In the context of IoT systems, choosing the optimal protocol is crucial, necessitating a comprehensive evaluation of the advantages and disadvantages of various protocols, such as HTTP, MQTT, and CoAP, considering factors such as massive data transmission, data loss rate, and stability. This study presents insights into the data transfer volume and loss rates for HTTP, MQTT, and CoAP. When implementing an IoT system, the choice of protocol is critical and must be made by considering the use case and requirements. In the context of this study’s AIoT system, choosing an MQTT protocol that can reliably handle significant volumes of data is an appropriate choice.

In this study, we introduced a model using CNN-based deep learning to determine the maturity of coffee trees in a smart farm environment. Thanks to the hierarchical structure of the CNN, it was possible to distinguish between the four maturity stages of coffee fruits (pinton, verde, rojo, sobremaduro) while maintaining high accuracy, even with images taken from various angles and distances. The model demonstrated excellent performance in evaluation metrics such as precision, recall, and IoU, proving that the pattern recognition capability of CNN is useful in farm environments. This confirms that the system can significantly contribute to determining harvest timing and quality control. This AI-based system has the potential to be applied not only to coffee farms but also to various smart agricultural environments. Due to CNN’s flexible feature-learning capabilities, it can be utilized in monitoring the growth stages of different crops, detecting pests and diseases, and predicting yields across multiple areas of agriculture. The introduction of such technology is expected to contribute to enhancing agricultural productivity and realizing sustainable farming, enabling real-time crop monitoring and rapid decision making, thus accelerating the realization of smart agriculture.

In summary, this study investigates the application of fog computing to alleviate bandwidth burdens in extensive IoT systems, facilitating the collection of substantial huge data. This data accumulation is vital for real-time monitoring of coffee fruit maturity level, underscoring the need for further research in automated environmental control systems for smart farming. Smart farm systems, by gathering real-time data on environmental variables, are expected to proficiently monitor and adjust crop growth conditions. Through the analysis of sensor data and subsequent environmental adjustments based on fruit maturity level, smart farms are poised to enhance water and nutrient management and climate control. This research is anticipated to boost productivity and promote efficient energy utilization.

Since this study focuses on the smart agriculture environment, particularly in coffee farms, the same performance cannot be assured in other industries. In large-scale agricultural environments, fog computing may present significant variables, such as scalability and implementation costs. These environments require more fog nodes and complex network infrastructures, which increase both initial setup and maintenance costs. Given that experimental verification has not been conducted in large-scale agricultural environments, it is essential to assess the applicability in more diverse conditions.

Additionally, security vulnerabilities can also pose significant challenges in large-scale environments. In a fog computing environment expanded on a large scale, more nodes and complex networks are required, which can lead to issues such as data integrity, access control, and network security. In particular, security can become vulnerable during data transmission across multiple nodes, necessitating further research into additional security mechanisms to address these vulnerabilities. Research aimed at resolving security issues must be considered essential when fog computing systems are scaled up, and this will contribute to enhancing the practical applicability of the architecture proposed in this study. Future research should explore the integration of this architecture with emerging technologies such as blockchain and 5G networks. Blockchain can enhance data security and transparency within the system, preventing sensor data tampering and ensuring traceability throughout the agricultural process. The 5G network can significantly improve real-time communication between IoT devices, fog nodes, and the cloud due to its ultra-low latency and high bandwidth capacity, thereby enhancing the system’s scalability and performance. Additionally, in urban environments like smart cities, fog computing could be applied in areas such as traffic management, energy consumption monitoring, and public safety management. By collecting and analyzing sensor data in real time, it can optimize traffic flow, improve energy efficiency, and monitor air quality across the city for rapid response. Fog computing can provide a flexible and resilient architecture for large-scale smart systems and support advanced AI models and predictive analytics. These research directions will ensure that the system is suited to future demands and capable of addressing the evolving requirements of smart agriculture.

## Figures and Tables

**Figure 1 sensors-24-06689-f001:**
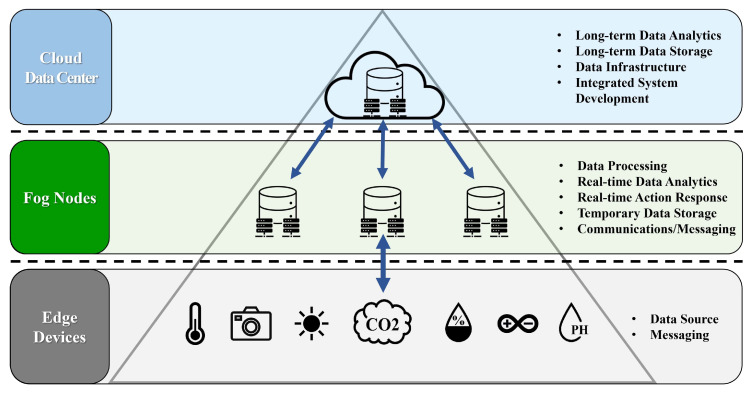
Fog computing architecture.

**Figure 2 sensors-24-06689-f002:**
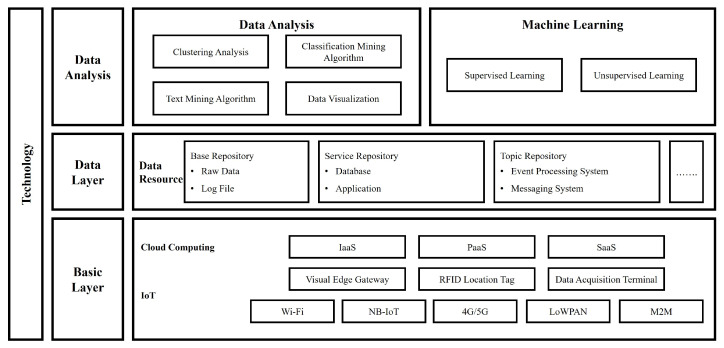
Technical framework of the application of AIoT.

**Figure 3 sensors-24-06689-f003:**
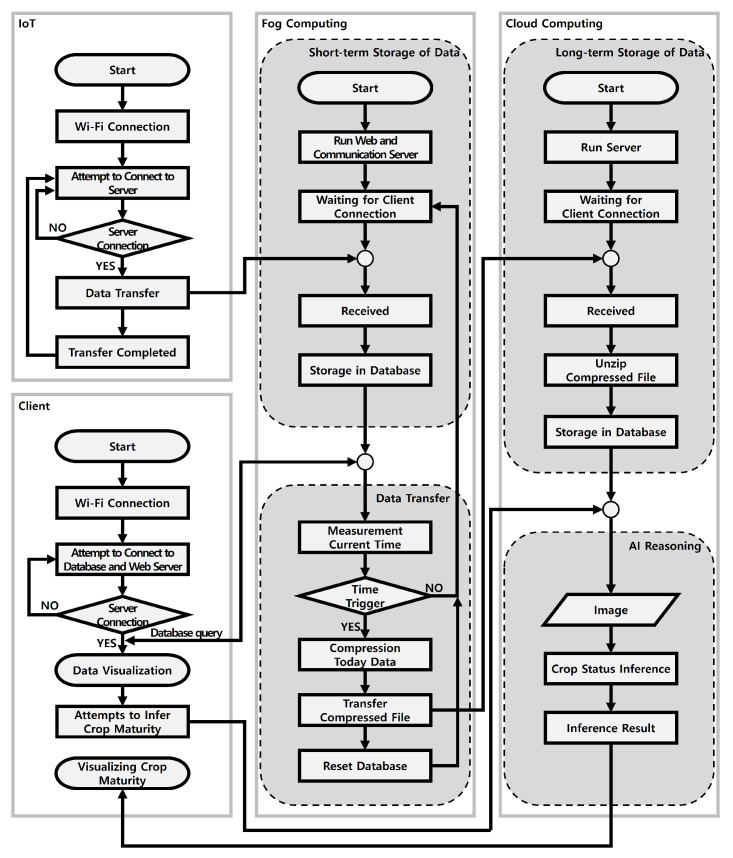
System architecture: smart farm system integrating fog computing.

**Figure 4 sensors-24-06689-f004:**
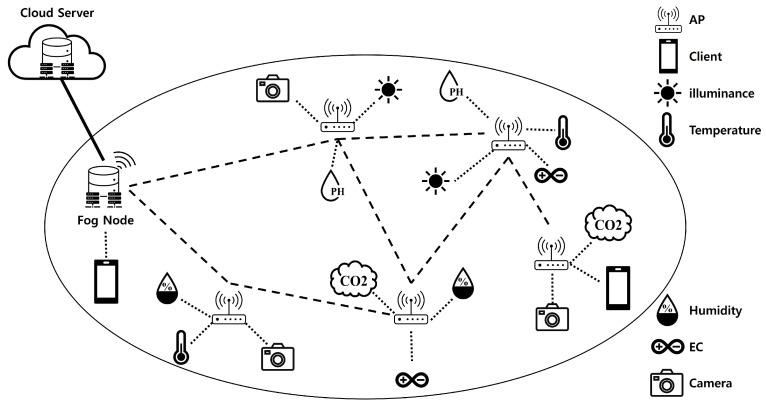
WMN in fog computing: optimizing IoT communication.

**Figure 5 sensors-24-06689-f005:**
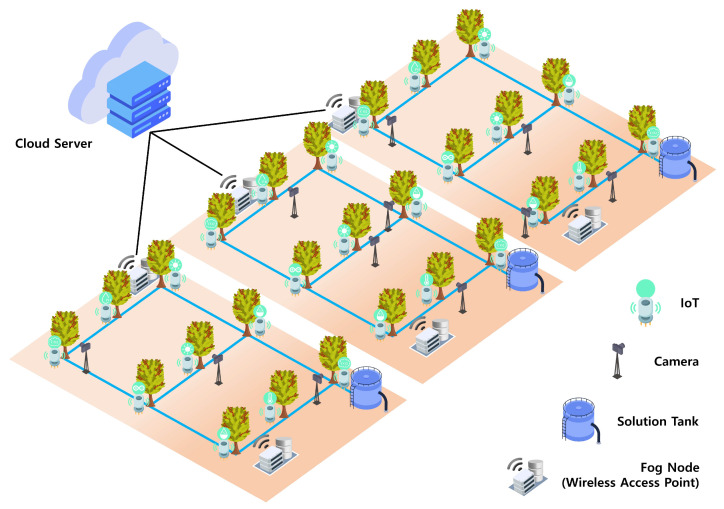
Fog computing-based smart farm environment at a hydroponic coffee farm.

**Figure 6 sensors-24-06689-f006:**
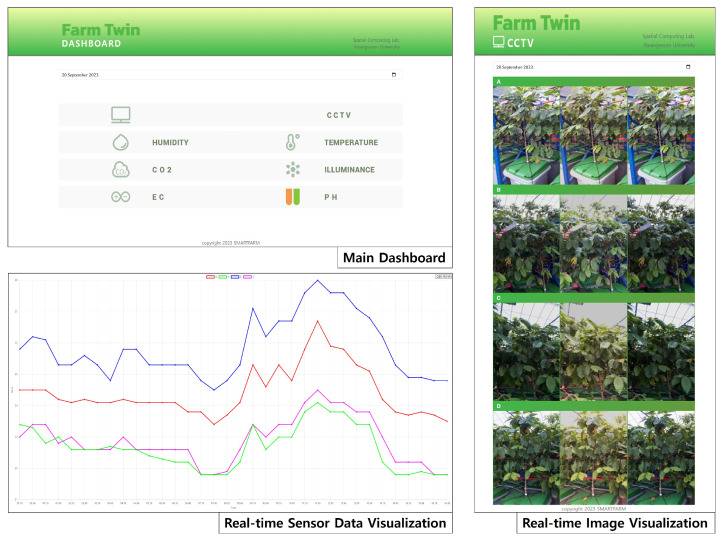
User interface of the web service page for the smart farm system.

**Figure 7 sensors-24-06689-f007:**
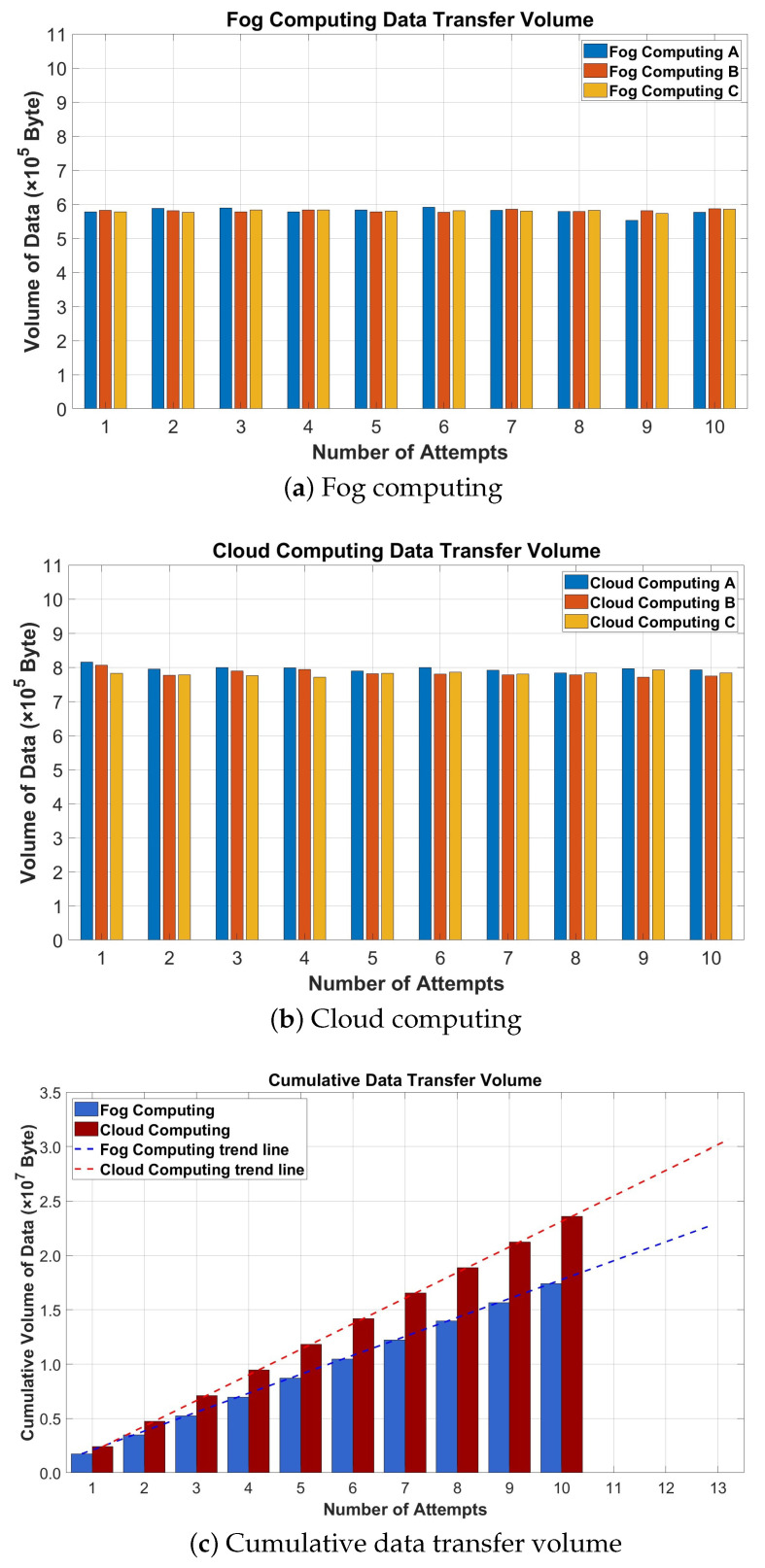
Fog computing/cloud computing data volume measurement results. (**a**) Data volume of fog computing. (**b**) Data volume of cloud computing. (**c**) Data volume of cumulative data transmission and forecast results.

**Figure 8 sensors-24-06689-f008:**
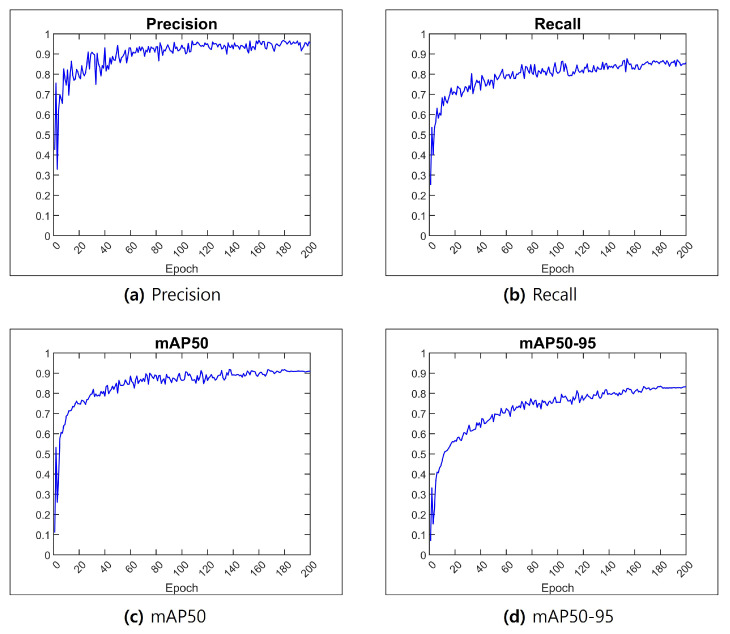
AI learning results. (**a**) Results for precision. (**b**) Results for recall. (**c**) Results for mAP50. (**d**) Results for mAP50-95.

**Figure 9 sensors-24-06689-f009:**
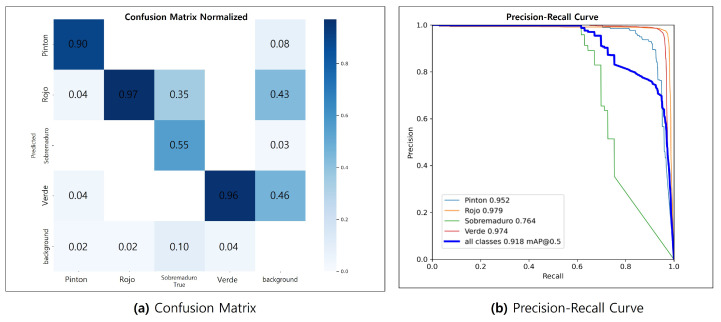
AI learning results. (**a**) Confusion matrix. (**b**) Precision–recall curve.

**Figure 10 sensors-24-06689-f010:**
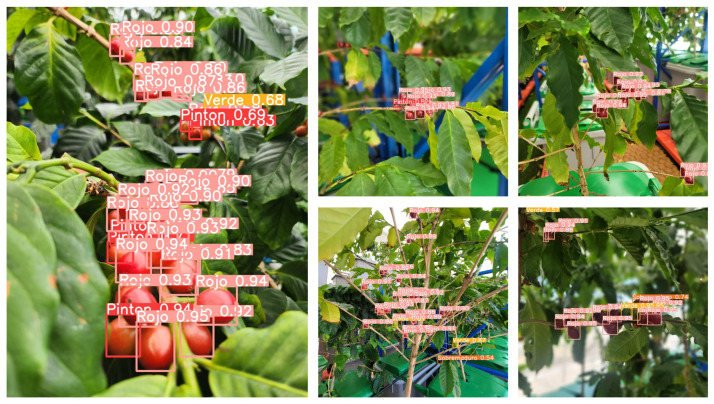
Coffee fruit AI inference results.

**Figure 11 sensors-24-06689-f011:**
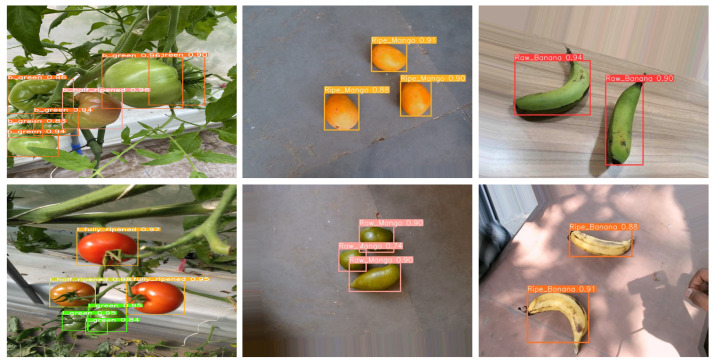
Fruit AI inference results.

**Table 2 sensors-24-06689-t002:** Structured database table: collected sensor data in fog nodes.

Field Name	Data Format	Key	Not Null
**ID**	Integer	P.K	O
**Sensor Group**	Text	-	O
**Sensor Name**	Text	-	O
**Sensor Data**	Text	-	O
**Data Time**	Text	-	X

**Table 3 sensors-24-06689-t003:** IoT device (WeMos D1 R1) environment.

Specification	Detail
**Microcontroller**	ESP-8266EX
**Operating Voltage**	3.3 V
**Digital I/O Pins**	11
**Analog Input Pins**	1 (Max input: 3.2 V)
**Clock Speed**	80 MHz/160 MHz
**Flash**	4 M bytes

**Table 4 sensors-24-06689-t004:** IoT device (camera sensor) environment.

Specification	Detail
**Sensor Model**	IMX219
**Resolution**	8 MP
**Active Pixels**	3280 × 2464
**Frame Rate (video mode)**	1080p47, 1640 × 1232p41, 640 × 480p206
**Data Format**	Raw Bayer 10 bit

**Table 5 sensors-24-06689-t005:** IoT devices (PH/EC/illuminance/${\mathrm{CO}}_{2}$ sensor) environments.

Specification	PH Sensor	EC Sensor	Illuminance	${\mathrm{CO}}_{2}$ Sensor
**Sensor Model**	SEN0161	DJS-1	HS-CDSM-ll	MH-Z19B
**Operating Voltage**	5.0 V	5.0 V	3.3–5 V	4.5–5 V
**Measuring Range**	pH 0–14	1–15 mS/cm	-	0–2000 ppm
**Measuring Temperature**	0–60 °C	0–40 °C	-	−10~50 °C
**Driving Ability**	-	-	Over 15 mA	-
**Output Type/Signal**	AO	AO	DO (0 and 1), AO	UART, PWM, DAC

**Table 6 sensors-24-06689-t006:** Fog node and cloud server environments.

Specification	Fog Node Server (VEEA VHE10)	Cloud Server (AWS EC2)
**Type**	-	t2.micro
**CPU**	ARMv8 quad-core processor, 1.5 GHz	vCPUs, 2.5 GHz
**RAM**	8 GB DRAM	1 GB
**Wi-Fi**	Tri-band Wi-Fi5	-
**Network**	1 Gbps	10 Gbps
**OS**	Ubuntu 20.04	Ubuntu 20.04
**DB**	SQLite 3.31.1	SQLite 3.31.1
**Web Server**	Apache 2.4.41	Apache 2.4.41
**Web Framework**	php 7.4.3	php 7.4.3

**Table 7 sensors-24-06689-t007:** AI server (AWS EC2) environment.

Specification	Detail
**Type**	c5d.large
**CPU**	vCPUs, 2 GHz
**RAM**	4 GB
**Network**	10 Gbps
**OS**	Deep learning AMI GPU Pytorch 2.0.1 (Ubuntu 20.04)
**Python**	3.8.18
**Opencv-python**	4.8.0
**Numpy**	1.24.3
**Psutil**	5.9.5
**Pillow**	10.0.1
**Scipy**	1.10.1
**Jinja2**	3.1.2
**Keras**	2.13.1
**Torch**	2.0.1
**Tensorflow**	2.13.0
**Ultralytics**	8.0.183

**Table 8 sensors-24-06689-t008:** AI learning settings and parameters.

Specification	Detail
**Model**	YOLOv8l
**Parameters (M)**	43.7
**FLOPS (B)**	165.2
**Epochs**	200
**Batch**	32
**Img_size**	640, 640
**Optimizer**	Adam
**Learning_rate**	0.001
**Momentum**	0.937

**Table 9 sensors-24-06689-t009:** IoT protocol data volume measurement results.

Protocol	Number of Sensors	1	2	3	4	5	6	7	8	9	10
CoAP	10 50 100	704 2992 6248	880 2904 6600	880 3608 7128	880 3168 6512	792 3080 7040	792 2992 6512	880 3608 5808	880 2728 8096	704 3344 7040	880 3872 6600
MQTT	10 50 100	870 4006 7658	760 3262 5662	870 3608 11,602	924 3642 11,602	760 3716 7822	924 3262 13,588	870 3910 7820	706 3856 7822	1218 3856 7464	514 3230 7768
HTTP	10 50 100	3870 16,343 42,156	3924 19,711 45,611	3978 21,024 43,794	3924 20,538 42,807	3924 20,646 41,779	4140 21,078 42,213	4086 20,376 41,294	4140 20,376 39,834	4032 20,484 39,186	4140 20,862 41,294

**Table 10 sensors-24-06689-t010:** IoT protocol loss rate measurement results.

Protocol	Number of Sensors	1	2	3	4	5	6	7	8	9	10
CoAP	10 50 100	60 32 30	10 34 25	0 18 20	0 28 27	10 30 20	20 36 26	0 26 34	0 40 8	20 26 20	0 12 15
MQTT	10 50 100	0 0 0	0 0 0	0 0 0	0 0 0	0 0 0	0 0 0	0 0 0	0 0 0	0 0 0	0 0 0
HTTP	10 50 100	0 0 0	0 0 0	0 0 0	0 0 0	0 0 2	0 0 1	0 0 0	0 0 0	0 0 0	0 0 0

**Table 11 sensors-24-06689-t011:** Performance measurement results of AI model.

Class	Precision	Recall	F1 Score	mAP50	mAP50-95
Coffee	0.952	0.858	0.902	0.910	0.832
Tomato	0.849	0.803	0.825	0.876	0.781
Banana and Mango	0.964	0.961	0.962	0.993	0.865

## Data Availability

The original contributions presented in the study are included in the article, further inquiries can be directed to the corresponding author.

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
