# Peer review of "Implementation of Smart Farm Systems Based on Fog Computing in Artificial Intelligence of Things Environments"

_sensors, 2024, doi:10.3390/s24206689_

Round 1
Reviewer 1 Report
Comments and Suggestions for Authors
The paper effectively identifies the limitations of traditional cloud computing in the context of IoT, especially the overhead caused by transmitting large volumes of data. The introduction of fog computing as a solution is well-justified and relevant in the context of modern smart farm systems. The study compares the network traffic, data loss rates, and performance of different IoT communication protocols (HTTP, MQTT, CoAP) in fog and cloud computing environments. This approach provides comprehensive insights into the impact of different protocols, and the introduction of a CNN-based algorithm for fruit maturity classification is both innovative and applicable to real-world agricultural challenges. The quantitative analysis presented in the results section, such as the data transmission volume and data reduction in fog computing, is clear and well-illustrated. The reduction in cumulative data volume (26% over ten days) is significant, demonstrating the effectiveness of fog computing in reducing overhead in IoT systems. The implementation of a CNN-based system to determine the maturity level of coffee fruits is a practical and forward-thinking application of AI in agriculture. The results of precision (0.947) and mAP (0.882) show strong model performance, making it a promising tool for farmers to monitor crop status more efficiently.
The paper seems interesting but it could benefit from some revisions to make it even better.
- While the technical and experimental details are thorough, the paper does not sufficiently address the potential limitations of the proposed fog computing system. For example, issues related to the scalability of fog nodes or the cost of implementing such systems in large-scale agricultural environments could have been discussed in greater detail.
- The paper mentions real-world experiments on coffee farms, but the broader generalizability of the results to other types of farms or IoT systems remains unclear. More experiments on diverse types of crops or in different climates could strengthen the claims of the research.
- The section on CNN-based maturity level classification provides good insights, but it could benefit from additional clarification on the data augmentation methods and why specific techniques (such as brightness adjustment and mosaic effects) were chosen. This would enhance the reproducibility of the AI model.
- Although the paper concludes with a mention of possible benefits for future applications, a more detailed exploration of next steps for research would be useful. For example, expanding on how the proposed fog-computing architecture could integrate with other emerging technologies like blockchain or 5G networks might enhance the forward-looking aspect of the research.
- Provide a more balanced discussion on both the strengths and limitations of fog computing in IoT systems. Address scalability, security concerns, and the potential for widespread adoption in different agricultural contexts.
- To strengthen the findings, consider expanding the experiments to different types of crops and farming systems. This will help validate the generalizability of the conclusions drawn in the paper.
- Explore how the system could be adapted for other applications outside of smart farming, such as in industrial IoT or urban environments, where fog computing might have different trade-offs or benefits.
Reviewer 2 Report
Comments and Suggestions for Authors
This study implements an artificial intelligence of things (AIoT) system based on fog-computing on a smart farm.
Tests are achieved on IoT messaging protocols (HTTP, MQTT, CoAP), and a CNN-based maturity level is performed to classify the images of coffee trees.
Some recommendations to enhance the quality of the study:
1- The introduction is limited (does not present the) and too short. You can for example: add the research questions, highlight the contributions (bullet points), describe the organization of the rest of sections.
2- Fig. 1 : too general representation of the architecture of the Fog computing. It will be better to customize the Fog architecture to your use case (Smart Farm AIoT based Systems).
3- The same previous recommendation for the AIoT system framework (Fig. 2).
4- The convolutional neural network architecture was not explained (input and output layers, number of hidden layers, weights, pooling, ..)
5- Globally, the explanation of the used methodology (section 3) is too general and simple.
6- Please add a figure legend to explain the symbols/icons in fig.4 (especially those referreing to humidity , temperature, and luminosity).
7- In the result (section 4), it is not indicated in the parameter setting if the values are taken based on empirical studies, on previous studies or on what basis? On the other hand, the values of the parameters of the architecture of the used convolutional neural network (number of hidden layers, ...) were not stated and justified.
8- Please revise the titles of all the tables and figures and make them more accurate (Example : Table 2. Database table : database of what? what the table contains in relation to this database).
9- I recommend to highlight the contribution of the CNN in a discussion subsection at the end of results. Moreover, Is it possible to replicate the achieved experiments: how can we reuse your work and findings? Please explain.
10- Please add perspectives and future research directions in the conclusion (based on the limitations of this current work).
Reviewer 3 Report
Comments and Suggestions for Authors
Major remarks:
While the introduction discusses IoT and fog computing, it would benefit from a stronger focus on smart farming applications. You make only a single mention of smart farms in the last lines of section 2, without providing any context on the subject. Establishing a clearer connection between IoT, fog computing, and their role in smart farming would enhance the introduction. Additionally, the objective of the paper is not clearly outlined. It would be helpful to introduce a concise statement summarizing the main goal of the study early in the introduction.
Furthermore, I noticed there was no mention of WebSocket in the document. WebSocket is particularly useful in scenarios where data from sensors, cameras, and other devices need to be processed instantly by fog nodes to make local decisions. WebSocket is full-duplex, continuous, and ideal for real-time communication, often established via HTTP. In my opinion, it would be beneficial to include a discussion of WebSocket in your work, given its relevance to fog computing environments, despite you didn’t use it.
The beginning of section 3 is very incomplete and does not allow the reader to easily understand how it fits into the farm context. It seems like a general architectural scheme for any fog computing system. You need to contextualize it better by explaining that, on a farm, specific types of data are generated, such as sensor data or camera feeds. These types of data are processed through an IoT layer, and then passed to the fog for real-time processing. In some cases, there is a need to access the cloud, while users may need to access both the fog node and the cloud itself. This explanation is essential to properly convey the system's function. Once these aspects are adequately explained, you should define the requirements for the project, or at least discuss the key components like the IoT Devices Layer, Wireless Mesh Network, Fog Computing Layer, and Cloud Computing Layer. Presenting an architecture without proper context and justification is not sufficient. After reading a well-structured description, the reader should be able to conclude that the system architecture is designed to balance real-time data collection and processing via fog computing, with the scalability and storage of the cloud. The requirements section should focus on optimizing local processing efficiency and long-term data management to ensure that smart farm operations run with minimal latency and bandwidth consumption. This can be followed by a detailed discussion of HTTP, MQTT, and CoAP protocols concerning this architecture.
For section 3.2, I assume that your IoT devices are connected to the fog nodes using HTTP, MQTT, or CoAP. In this section, I expected the authors to include more details about the modular architecture of the fog node, explaining how it is structured to handle the different tasks listed in subsections 3.2.1, 3.2.2, and 3.2.3. According to Figure 3, and considering the absence of clear requirements, one of the key features of the fog nodes is providing real-time data visualization to the farmers via the web server (you will discuss it later in the paper). However, it is unclear how this is achieved using MQTT, for example, and this should be clarified. Lastly, you should also address the energy efficiency aspect of the fog nodes. Given that smart farms often involve resource-constrained devices, it would be important to explain how fog computing reduces the energy load on these IoT devices by offloading computational tasks to the fog nodes.
Minor remarks:
The sentence: “This study analyzes the effectiveness of fog-computing in the context of IoT.” For me it is missing that your study is in the smart farm context.
When you explain your approach in lines 40-42, I recommend a figure to explain better the joint IoT-fog smart farm.
When you say “In a fog-computing environment, IoT devices operate within the confines of a local area network (LAN), bypassing the need to traverse a wide area network (WAN), reducing the network-related variables.”, this is inaccurate because while fog computing reduces the dependency on a WAN for some operations, it does not entirely bypass the WAN in some cases. Many IoT environments still require a cloud connection for more significant data processing or storage, which involves WAN communication. Thus, while fog computing minimizes the need for constant WAN traversal, it doesn't eliminate it. Note that you will have a cloud connection in Figure 3.
The way Table 1 is presented in the text is abrupt, it would be better to discuss that HTTP, MQTT, and CoAP are important protocols and then present the table.
In my opinion, Figure 2 as presented in the document is not useful. Firstly, the figure is poorly designed, with boxes classifying different aspects like data layers, analysis layers, and connection layers, all together, which does not give the reader any clear perspective. Secondly, your explanation of the figure does not add clarity—it aligns with my lack of understanding of it. I recommend that you either provide a much clearer explanation of this AIoT context or remove the figure, as it currently adds little to the overall value of the work.
Line 110: This(?)
Line 180: The way the WMN appears in the text out of nowhere, it's not easy for the reader to follow.
In section 3.1, a table about the characteristics of HTTP, MQTT, and CoAP would be important.
n Section 3.3.2, critical training parameters such as the learning rate, batch size, and number of epochs are missing. Additionally, the choice of optimizer (e.g., Adam, SGD) should be specified. It is also necessary to explain how cross-validation or test set evaluation was performed. The metrics used to evaluate the model, such as accuracy, precision, recall, and F1 score, should be clearly defined. Moreover, I recommend including examples of confusion matrices or precision-recall curves to demonstrate how well the model differentiates between stages like "Rojo" and "Sobremaduro." The current explanation lacks detail about the modeling process, as the authors only included a figure with results. The process leading to these results should be properly documented.
Figure 5, in my opinion, this figure should have appeared much earlier in the paper when you were establishing the requirements and proposing the architecture of the system in Figure 3. Introducing it earlier would provide a clearer context for the system design and enhance the reader's understanding of how the architecture is structured.
Line 276-279: There should also be a table for sensors
Linha 284: Only users inside the farm can view this, right, because of the fog node? If they're already there, why do they need this access?
Line 285: How the UI was developed?
The font size on the axes of Figures 7 and 10 is so small that it's impossible to read.
The explanation of section 5.2 is almost non-existent, it needs to be substantially improved, perhaps presenting the results in tabular form would be more appropriate
Round 2
Reviewer 1 Report
Comments and Suggestions for Authors
The CNN-based classification section could benefit from more detailed information regarding the chosen data augmentation techniques (brightness adjustment and mosaic effect). While the revised version provides some explanation of data augmentation, the reasoning for selecting these specific techniques remains insufficient. It would be helpful if the authors could elaborate on how these techniques impact the model's generalizability and robustness in real-world scenarios
Although the revised version does present a more balanced view of fog computing, it still does not fully address potential security issues. Security vulnerabilities in fog computing, particularly in large-scale implementations, should be addressed in greater depth
The revised version does briefly mention scalability and cost, but it remains too short and lacks depth. It is still necessary to explore in more detail how these systems can scale in large agricultural operations, including the potential need for more fog nodes and the associated costs.
While the revision has added a section on applying this system to other crops (e.g., tomatoes and bananas), it could still be more specific in explaining whether the same fog computing advantages (such as data reduction and real-time decision-making) could be applied in environments with different climates, soil conditions, or crop requirements.
Reviewer 2 Report
Comments and Suggestions for Authors
Please add the access time of the added web references [39] and [40].
Reviewer 3 Report
Comments and Suggestions for Authors
Congratulations to the authors for the work
Author Response
Thank you very much for your kind words and positive feedback. We truly appreciate your support and are grateful for the opportunity to improve our work based on the valuable suggestions provided.